# Role of MicroRNA-502-3p in Human Diseases

**DOI:** 10.3390/ph16040532

**Published:** 2023-04-02

**Authors:** Davin Devara, Yashmit Choudhary, Subodh Kumar

**Affiliations:** 1 Center of Emphasis in Neuroscience, Department of Molecular and Translational Medicine, Paul L. Foster School of Medicine, Texas Tech University Health Sciences Center, El Paso, TX 79905, USA; 2 Maxine L. Silva Health Magnet High School, 121 Val Verde St., El Paso, TX 79905, USA; 3 L. Frederick Francis Graduate School of Biomedical Sciences, Texas Tech University Health Sciences Center, El Paso, TX 79905, USA

**Keywords:** Alzheimer’s disease, microRNAs, MiR-502-3p, synapse, GABAergic neuron

## Abstract

MicroRNAs (miRNAs) are non-coding RNAs that play a major role in gene regulation in several diseases. MicroRNA-502-3p (MiR-502-3p) has been previously characterized in a variety of human diseases such as osteoporosis, diabetes, tuberculosis, cancers, and neurological disorders. Our studies recently explored the new role of miR-502-3p in regulating synapse function in Alzheimer’s disease (AD). AD is the most common cause of dementia in elderly individuals. Synapse is the initial target that is hit during AD progression. The most common causes of synapse dysfunction in AD are amyloid beta, hyperphosphorylated tau, and microglia activation. MiR-502-3p was found to be localized and overexpressed in the AD synapses. Overexpression of miR-502-3p was correlated with AD severity in terms of Braak stages. Studies have shown that miR-502-3p modulates the glutaminergic and GABAergic synapse function in AD. The current study’s emphasis is to discuss the in-depth roles of miR-502-3p in human diseases and AD and the future possibilities concerning miR-502-3p as a therapeutic for AD treatment.

## 1. Introduction

Alzheimer’s disease (AD) is a progressive, neurodegenerative disorder that affects about 6.5 million people in the United States (US) and 50 million people worldwide [1,2]. In the US, it is the most common cause of dementia and the seventh leading cause of death [3]. In total, 10.7% of people over the age of 65 have AD in the US. In the next 35 years, the world’s older population will outpace the younger population. By 2050, the population aged 65 and older will make up about 15.6% of the world’s population [4]. This will cause a dramatic increase in AD cases, projected to be around 12.7 million people in the US by 2050 [3]. There is currently no definitive cure for AD.

AD commonly presents clinically with memory impairment, language impairment, disorientation, and impairment of judgment [5]. The symptoms are a result of neurodegeneration believed to be caused by amyloid beta (Aβ) plaques and neurofibrillary tangles (NFTs). The Aβ peptides from the amyloid β precursor protein (AβPP) make up the Aβ plaques and hyperphosphorylated tau (p-tau) proteins comprise the NFTs. Deposition of Aβ plaques and NFTs in cholinergic neurons cause proinflammatory events that lead to neurodegeneration, which deteriorates cognition [6].

Early-onset Alzheimer’s Disease (EOAD) is defined as AD occurring before the age of 65 [7]. Amyloid protein precursor (APP), presenilin-1 (PSEN1), and presenilin-2 (PSEN2) are all genes that affect amyloid production and cleavage. Mutations in these genes have been found to lead to 5–10% of EOAD cases [8].

Late-onset Alzheimer’s Disease (LOAD) is multifactorial, meaning that effects from multiple genes and environmental factors interact together to produce the disease. Over 20 gene loci have been identified to contribute to LOAD, with genotype E4 of the Apolipoprotein E (APOE) gene being one of the major risk factors [9]. Additional, non-genetic risk factors include obesity, type 2 diabetes, lack of exercise, high-sugar diets, and stress [10]. Age is the most critical risk factor that contributes to both EOAD and LOAD [11].

Currently, the only accurate way to diagnose AD is by brain autopsy. Other methods include positron emission tomography and analysis of cerebrospinal fluid for p-tau, Aβ42, and total tau protein content [6]. However, to successfully diagnose AD, the disease would need to have already progressed to a relatively advanced stage. There has been a considerable amount of research in recent years to try to find ways to detect AD early and hopefully treat it. One of the potential kinds of biomarkers being explored is microRNA (miRNA). In this review, we have highlighted the roles of microRNA-502-3p (miR-502-3p) in human diseases with a particular emphasis on AD.

## 2. MicroRNAs

MiRNAs are single-stranded, non-coding RNAs that play a major role in gene regulation. In many cases, miRNAs suppress gene expression by interacting with the 3′UTR region of their target messenger RNAs (mRNAs); however, miRNAs interacting with 5′ UTR, coding sequence, and gene promoters have also been reported [12]. Some miRNAs that interact with gene promoters have been found to induce gene expression. For example, miR-373 targets the E-cadherin promoter to increase gene expression [13]. MiRNAs are involved in most cellular processes, which include viral replication, immune responses, aging, cholesterol metabolism, insulin secretion, neurogenesis, cardiac and skeletal muscle development, hematopoiesis, stem cell differentiation, developmental processes, apoptosis, and cell-cycle control [14,15,16,17].

According to the most recent version (v22.1) of the miRBase, 38,589 hairpin precursors and 48,860 mature miRNA sequences have been identified across 271 species [16]. MiRNA biogenesis starts in the nucleus and finishes in the cytoplasm [16]. In the nucleus, RNA polymerase II or RNA polymerase III transcribes a large primary-RNA transcript (pri-RNA) from a gene [11]. Pri-RNA is further cleaved into an ~85 nucleotide stem-loop structure called precursor RNA (pre-RNA) by DROSHA/DGCR8 protein complex [18]. Afterward, pre-RNA is transported into the cytoplasm by a transporter containing Exportin 5 and Ran-GTP [19]. Dicer, an RNase type III, cleaves pre-RNA into a ~20–22 nucleotide duplex, with one strand being the mature miRNA strand and the other one being the complementary passenger strand. After the duplex is unwound, the mature miRNA strand is incorporated into an RNA-induced silencing complex (RISC), guiding RISC to the target mRNA [18].

Because miRNAs are a part of many cellular processes, their dysregulation has been associated with many diseases. With this in mind, miRNAs serve as potential biomarkers and therapeutic targets. Research has shown that different miRNAs can be used as biomarkers for multiple types of cancers, viral infections, cardiovascular disorders, diabetes and other metabolic disorders, and nervous system disorders [20]. There have also been many therapies associated with miRNA targets. For example, a mimic of tumor suppressor miRNA miR-34 has been developed for cancer treatment and an anti-miR for miR-122 has been developed for hepatitis treatment [21].

As mentioned before, several miRNAs have been linked to AD pathogenesis. A meta-analysis in 2019 found a significant differential expression of 57 miRNAs in the brain (25 miRNAs), cerebrospinal fluid (CSF) (5 miRNAs), blood-derived specimens (32 miRNAs), and both brain and blood (5 miRNAs) [22]. In 2016, we reviewed miRNA biomarkers related to AD [23]. Several miRNAs have been identified as targets for AD therapy. One example is miR-485-3p, which is found to be overexpressed in the brain tissues, CSF, and plasma of AD patients. A miR-485-3p antisense oligonucleotide treatment was found to reduce Aβ plaque accumulation, tau pathology development, neuroinflammation, and cognitive decline in mice [24]. Our past studies explored the biomarker and therapeutic potential of miR-455-3p in AD [25,26,27,28]. Recently, we found novel evidence of the role of miR-501-3p and miR-502-3p in synapses of AD brains, which could serve as a potential biomarker and therapeutic target in the future [29]. Our lab is mostly investigating the role of miRNAs in AD, and as mentioned earlier, we identified miR-501-3p and miR-502-3p as the main targets in AD brain synapses. This review aims to understand the roles of miRNA-502-3p in other human diseases to unveil how its molecular mechanisms differ or are similar to those in AD.

## 3. MiR-500 Family

The microRNA-500’s family has five different genotypes: microRNA-362, microRNA-500a, microRNA-500b, microRNA-501, and microRNA-502 (Genesnames.org). All the forms of miR-500 family members are expressed in humans and different animal species. The miR-502-3p sequence is 22 nucleotides long and is found in Homo sapiens (Has-miR-502-3p) as annotated by 7 gene databases such as MalaCards, miRBase, GeneCards, TarBase, ENA, RefSeq, and LncBase. The miR-502 were also found to be conserved in seven different animal species such as Cow (*Bos taurus*) Bta-miR-502a; Dog (*Canis lupus familiaris*) Cfa-miR-502; Horse (*Equus caballus*) Eca-miR-502-3p; Gorilla (*Gorilla gorilla*) Ggo-miR-502a; Rhesus monkey (*Macaca mulatta*) Mml-miR-502-3p); Rabbit (*Oryctolagus cuniculus*) Ocu-miR-502-3p; and Bornean orangutan (*Pongo pygmaeus*) Ppy-miR-502-3p (https://rnacentral.org/rna) (accessed on 26 February 2023). The miR-502-3p is encoded by the MIR502 gene (ENSG00000272080) which is composed of an 86 base-pairs genomic sequence, a plus stranded RNA orientation starting from 50,014,598, and ending at 50,014,683. The MiR502 gene is located at the Chromosome X genomic location: 50,014,598-50,014,683 forward strands. The MIR502 gene has one transcript (splice variant), fifty-eight orthologues, and three paralogue components (www.useast.ensembl.org) (accessed on 26 February 2023). This MIR-502-3p transcript also has one exon which is associated with thirty-nine variant alleles, and maps to fifty-seven oligo probes. The hsa-miR-502-3p sequence is a product of hsa-miR-502, miR-502-3p, MIR502, miR-502, and hsa-miR-502-3p genes. Figure 1 shows the secondary structure of MIR-500 with percentage (%) of G-C contents. The hsa-miR-502-3p was reported to interact with several protein-coding genes, including 156DAG, 182-FIP, 2700066J21Rik, 2H9, 80K-L, 82-FIP, A2MR, A3a, AAMP, and ABCA12 (https://rnacentral.org/rna/URS0000601CC4/9606/) (accessed on 26 February 2023). In humans, miR-502 is more heavily expressed in the following tissues: subcutaneous adipose, aorta, the cerebellar hemisphere of the brain, spinal cord, cultured fibroblast cells, muscularis esophagus, left ventricle of the heart, liver, and lung (https://www.gtexportal.org/home/gene/MIR502/) (accessed on 6 February 2023). MiR-501-3p and miR-502-3p are very close members with a single base sequence difference, however their seed sequence is similar. We analyzed the predicted targets of miR-501-3p and miR-502-3p using TargetScan, TargetMiner, and miRDB. The top 100 predicted targets are listed in Appendix A.

## 4. MiR-502-3p and Human Diseases

The first instance of miR-502-3p associated with disease came in 2014 when it was found to be upregulated in Merkel Cell Carcinoma [30]. Since then, further research has found miR-502-3p to be implicated in other human diseases such as osteoporosis, diabetes, muscular dystrophy, tuberculosis, multiple cancers, and neurodegenerative disorders (Table 1). Figure 2 depicts the role of miR-502-3p in most common human diseases.

### 4.1. Osteoporotic Fractures

Osteoporosis is a condition of decreased bone mass, which leads to diminished bone strength and predisposes patients to osteoporotic fractures [31]. In osteoporotic fractures, increased proinflammatory cytokines, such as TNF-α, play a role in bone loss [32]. Zhang et al. (2021) studied the correlation between TNF-α, omentin-1, and miR-502-3p in patients with osteoporosis [33]. Omentin-1 is a recently discovered adipocytokine that displays anti-inflammatory effects [33]. In their study, they found that omentin-1 and miR-502-3p were both underexpressed in osteoporosis patients with osteoporotic fractures compared to osteoporosis patients without osteoporotic fractures. Furthermore, TNF-α was independently, negatively correlated with both omentin-1 and miR-502-3p. Omentin-1 and miR-502-3p were also found to be positively correlated with each other. Overall, the study confirmed the predictive value of omentin-1 and miR-502-3p in the diagnosis and prognosis of osteoporotic fractures [33]. The correlation between miR-502-3p, omentin-1, and other osteoporotic fractures must still be analyzed in future studies.

### 4.2. Diabetes

Diabetes mellitus (DM) is a disease that affects 37.3 million people in the US, which is 11.3% of the US population [34]. DM is an issue related to insulin, which ultimately interferes with the body’s ability to uptake glucose and leads to hyperglycemia [35]. It is commonly categorized into two types: Type 1 DM (T1DM) and Type 2 DM (T2DM) [36]. T2DM is primarily caused by defective insulin secretion by pancreatic β-cells and the inability of insulin-sensitive tissues to respond to insulin [37].

Montastier et al. (2019) explored the role of miR-502-3p in insulin resistance, which is one of the primary causes of T2DM [38]. Niacin is an anti-dyslipidemic drug that is known to cause insulin resistance. In the study, obese men were treated with either niacin, acipimox (a niacin-like drug that does not induce insulin resistance), or transfected with miR-502-3p. The biopsy of subjects’ adipose tissues showed that miR-502-3p was upregulated in the subjects who received Niacin treatment. There was no significant change in the expression of miR-502-3p in the acipimox treatment group. The study found that the overexpression of miR-502-3p caused impaired insulin-stimulated glucose uptake. This suggests that niacin-induced insulin resistance is mediated by miR-502-3p, leading one to believe that anti-miR-502-3p could be helpful in treating obesity-related insulin resistance [38].

### 4.3. Tuberculosis 

Tuberculosis (TB) is an infectious disease caused by the bacteria *Mycobacterium tuberculosis (M. tuberculosis)* that most often affects the lungs. It is highly prevalent in the socioeconomically poor populations of the world [39]. Liu et al. (2020) found that miR-502-3p plays a role in TB by facilitating the survival of *M. tuberculosis* in macrophages. Infections are combated with the help of proinflammatory cytokines such as IL-6, TNF-α, and IL-1β. These cytokines were significantly reduced in *M. tuberculosis*-infected macrophages when transfected with a miR-502-3p mimic. Rho-associated coiled-coil-forming protein kinase 1 (ROCK1) acts to activate monocyte proinflammatory response via the TLR4/NF-κB pathway and is a direct target of miR-502-3p via a receptor in the 3′-UTR of ROCK1. MiR-502-3p decreases the protein expression of ROCK1, which downregulates the TLR4/NFκB pathway and leads to decreased proinflammatory cytokine response and promotes *M. tuberculosis* survival [40].

### 4.4. Other Human Diseases

MiR-502-3p has also been found to be associated with several other diseases; however, these studies did not focus on miR-502-3p. One example is that miR-502-3p was found to be upregulated in lamin A/C-associated muscular dystrophy, but further studies must be completed to determine the specific role it plays in disease pathology [41]. A study was also conducted to analyze the expression of miR-502-3p in polycystic ovarian syndrome (PCOS). They reported that there was no significant upregulation of miR-502-3p in PCOS patients, but the authors acknowledged that this could be because all subjects were still in the early stage of PCOS and that miR-502-3p may play a role in the progression of the disease [42]. Future studies must be completed involving patients in different stages of PCOS progression.

## 5. MiR-502-3p and Human Cancers

MiR-502-3p has been found to be associated with many human cancers. These include triple negative breast cancer, follicular lymphoma, chronic lymphocytic leukemia, invasive pituitary adenoma, colorectal cancer, esophageal cancer, gallbladder cancer, hepatic cancer, lung cancer, pancreatic cancer, renal cancer, skin cancer, and stomach cancer. Here, we discuss findings of the molecular mechanisms of miR-502-3p as it relates to the listed cancers (Table 1). Figure 3 shows the role of miR-502-3p in most common human cancers.

### 5.1. Chronic Lymphocytic Leukemia

Chronic lymphocytic leukemia (CLL) is the most common leukemia in developed countries and mostly presents in the elderly, with the median age of diagnosis being 72 years old [43]. Ruiz-Lafuente et al. (2015) studied miRNAs associated as part of the interleukin-4 (IL-4) pathway in CLL. IL-4 promotes B cell differentiation and protects CLL B cells from apoptosis [44]. This study observed the upregulation of miR-21 and the miRNAs hosted within the CLCN5 gene, including miR-502-3p, when IL-4 was stimulated. It is possible that the miRNAs upregulated by IL-4 stimulation play a role in the anti-apoptotic effects of the IL-4 pathway in CLL, but further studies are required. If proven, miR-502-3p, along with the other miRNAs upregulated by IL-4, could be a promising target for CLL therapy [45].

### 5.2. Invasive Pituitary Adenoma

Pituitary adenomas are benign neoplasms that make up 10–15% of all intracranial masses [46]. About 30% of pituitary adenomas exhibit invasive behaviors and are classified as invasive pituitary adenomas (IPA) [47]. The role of miR-502-3p in IPAs was explored by Li et al. (2021). Long noncoding RNAs (lncRNAs) are transcripts with over 200 nucleotides but lack open reading frames [48]. Li et al. (2021) found that LINC00473 was the most upregulated lncRNA in IPA and that it downregulates miR-502-3p. Overexpression of LINC00473 promotes cell proliferation while overexpression of miR-502-3p inhibits cell proliferation in pituitary adenomas. Furthermore, the study determined that KMT5A is the target gene of miR-502-3p. KMT5A is a methyltransferase that increases the expression of cyclin D1 and CDK, promoting cell proliferation in IPA [47]. Upregulation of miR-502-3p was shown to inhibit the expression of KMT5A, which suggests that miR-502-3p is protective in IPA. 

### 5.3. Colorectal Cancer

Colorectal cancer accounts for the second most cancer deaths in the United States when considering both men and women [49]. Colorectal intramucosal cancer (IMC) is defined as carcinoma confined to the mucosal layer of the colon [50]. A genome-wide analysis of microRNA and mRNA expression in colorectal cancer uncovered an association between IMC and miR-502-3p [51]. The study found that miR-502-3p was inversely correlated with olfactomedin4 (OLFM4). OLFM4 is a glycoprotein expressed in intestinal crypts that functions to preserve intestinal Lrg5-positive stem cells. miR-502-3p was found to regulate OLFM4. Overexpression of miR-502-3p causes a reduced expression of OLFM4, which could be associated with IMC pathology. 

### 5.4. Gallbladder Cancer

Gallbladder cancer (GBC) is an extremely aggressive malignancy with a 5-year survival rate of less than 20% [52]. Hu et al. (2019) found a lncRNA highly expressed in GBC (lncRNA-HGBC) that is upregulated in GBC and asserts its effects through an axis involving miR-502-3p. LncRNA-HGBC binds to HuR, which stabilizes the lncRNA. Then, LncRNA-HGBC directly binds to and inhibits miR-502-3p. This sequestration of miR-502-3p causes an upregulation of SET, a proto-oncogene. SET causes the downstream activation of AKT, which contributes to the proliferation of GBC cells. This study revealed that the lncRNA-HGBC/miR-502-3p/SET/AKT axis is important in GBC progression and proposes that lncRNA-HGBC could be a potential therapeutic target in the future [53]. 

### 5.5. Lung Cancer

Interstitial lung abnormalities (ILA) are interstitial patterns incidentally detected on CT scans in patients without previously known lung diseases [54]. ILA can progress to multiple lung diseases, including interstitial pulmonary fibrosis and lung cancer [55]. Patients with ILA were found to have an increased risk for lung cancer and lung cancer mortality [56]. Ortiz-Quintero et al. (2020) found that miR-502-3p is upregulated in ILA and can be used as a biomarker to predict the presence of ILA in asymptomatic patients with a sensitivity of 51% and a specificity of 81%. KEGG pathway analysis found that the p53 signaling pathway is affected by miR-502-3p [55]. 

Lung cancer is the most commonly diagnosed cancer and accounts for the most cancer-related deaths, globally [57]. Lung cancer is classified as either small-cell lung cancer or non-small cell lung cancer (NSCLC). Lung adenocarcinoma is the most prevalent type of NSCLC [58]. In EGFR-mutated lung adenocarcinoma, miR-502-3p was found to be upregulated. Pathway analysis showed that miR-502-3p, miR-500a-3p, and miR-652-3p (two other upregulated miRNAs in EGFR-mutated lung adenocarcinoma), interact with and downregulate MUC4. Downregulated MUC4 has previously been shown to promote tumor progression in EGFR-mutated lung adenocarcinoma [59]. Additionally, miR-502-3p was found to be overexpressed in lung adenocarcinoma when compared to lung squamous cell carcinoma, another type of non-small cell lung cancer [60]. However, the differences in miRNA-502-3p expression between the two types of cancers and how it affects their respective pathophysiologies have yet to be explored. 

The prognosis of lung cancer has also been related to miRNA expression. Zhang et al. (2019) explored the diagnostic value of extracellular vesicles (EVs)-derived miRNAs for patients with pulmonary ground-glass nodules (GGNs). These nodules can be benign or malignant, but they are difficult to differentiate through biopsy. This study introduced a new method to potentially diagnose malignancies using plasma miRNAs derived from EVs. While designing the diagnostic support vector-machine model, they also found that miR-502-3p, along with miR-500a-3p and miR-501-3p, were associated with the progression of tumor cells. Patients with upregulated expression of these three miRNAs showed enhanced overall survival. Furthermore, when comparing miRNA expression before and after surgery, they found that miR-502-3p, miR-500a-3p, and miR-501-3p were all upregulated post-surgery. While the molecular connection between tumor progression and these three miRNAs must be further explored, it is clear that these miRNAs could possibly aid physicians in creating individualized treatment plans concerning lung cancer [61]. 

### 5.6. Pancreatic Cancer

Pancreatic cancer has one of the lowest five-year survival rates (11%) because it is often diagnosed at an advanced stage [62,63]. Tan et al. (2018) found that miR-502-3p was downregulated in pancreatic cancer. They predicted that miR-502-3p targeted the following mRNAs: KCTD9, RNF144A, DOK6, PTPRF, PDE3B, RORA, MYCN, DAPK1, ADAMTS3, CBLL1, and RBMS1. Further KEGG analysis shows that miR-502-3p and the other downregulated miRNAs in pancreatic cancer may be associated with tight junctions and apoptosis, possibly preventing tumor invasion and metastasis. While more studies must still be completed, these findings suggest that miR-502-3p plays some role in pancreatic cancer [64].

### 5.7. Stomach Cancer

In the US, it was estimated that there would be over 26,000 new cases of gastric cancer (GC) and over 11,000 deaths due to GC [65]. Recent works have uncovered the role of microRNA in GC. Li et al. (2020) found that miR-502-3p was part of a regulatory axis that is associated with gastric cancer: circ-RPL15/miR-502-3p/OLFM4/STAT3. Circular RNAs (circRNAs) are noncoding RNAs that are involved in the pathogenesis of many cancers. Circ-RPL15, a circRNA, promotes GC progression by functioning as a competitive endogenous RNA and sponging miR-502-3p. MiR-502-3p normally downregulates OLFM4 and p-STAT3 (phosphorylated-STAT3). OLFM4 is a protein with anti-apoptotic effects that also promotes tumor progression by enhancing STAT3 activation. These findings suggest that miR-502-3p has anti-tumor effects in GC [66]. This is further supported when Kim et al. (2021) found that downregulated miR-502-3p was associated with lymph node metastasis in intramucosal gastric cancer [67]. 

### 5.8. Other Cancers

Across the literature relating to miR-502-3p, abnormalities in its expression were also found to be associated with various cancers; however, these studies did not focus on miR-502-3p, so they were not discussed in this review. MiR-502-3p was found to be upregulated in Merkel cell carcinoma [30], conjunctival malignant melanoma [68], triple negative breast cancer [69], and follicular lymphoma [70]. The mechanisms by which miR-502-3p upregulation impacts these malignancies must be further explored. Liu et al. (2022) also found that miR-502-3p, along with six other miRNAs, may be critical in the live tumor progression of hepatocellular carcinoma [71]. Additionally, differences in miR-502-3p expression were found to be essential in distinguishing between normal tissue and renal cell carcinoma tissues [72]. Finally, miR-502-3p was found to be downregulated in Barrett’s carcinogenesis, an esophageal adenocarcinoma [73].

## 6. MiR-502-3p and Neurodegenerative Disease

There is a very limited amount of research that exists concerning miR-502-3p and neurodegenerative disorders. Current literature links miR-502-3p to different types of dementia, including AD, frontotemporal dementia (FTD), and vascular dementia (VD). Until our laboratory began to explore the role of miR-502-3p in AD, it was known to be a potential biomarker across different forms of dementia. When combined with analyzing the expression of miR-633a and miR-206, miR-502-3p can be used as a biomarker to diagnose FTD with a 100% sensitivity and 87.5% specificity. Global miRNA profiling showed downregulation of all three of these miRNAs in FTD [74]. MiR-502-3p can also be used as a potential biomarker in VD caused by cerebral small vessel disease with a 75% sensitivity and 89% specificity. It was found to be upregulated by 4.68-fold in small vessel VD [75]. Finally, Satoh et al. (2015) identified miR-502-3p as a potential biomarker in AD after finding that it is downregulated in the plasma of AD patients. The study further discussed how all 14 potentially downregulated miRNAs, including miR-502-3p, in AD are related to neuronal synaptic functions [76]. The specific role of miR-502-3p in the pathology of the previously mentioned dementias must be further explored. 

## 7. MiR-501-3p and Neurodegenerative Disease

While the literature on miR-502-3p and neurodegenerative disorders remains limited, miR-501-3p, another miRNA we recently characterized in AD, seems to have a more explored role in neurological functions. Our study found the significant upregulation of both miR-502-3p and miR-501-3p in the AD synapse [29]. Since miR-501-3p and miR-502-3p belong to the same family and the seed sequence of both miRNAs is similar in humans, it is important to explore the role of miR-501-3p in neurological disorders including AD. 

### 7.1. Schizophrenia

Schizophrenia is a complex psychiatric disorder that presents with positive (hallucinations, delusions) symptoms, negative symptoms (lack of interest, lack of concentration), and cognitive dysfunction and affects 32% of people worldwide [77,78]. Liang et al. (2022) investigated the role of miR-501-3p in schizophrenia. They found that miR-501-3p was downregulated in schizophrenia. Using miR-501-3p knockout male mice, the study showed that the loss of miR-501-3p resulted in the loss of total dendritic spine density, suggesting that the miRNA is necessary for regulating synaptic structure or function. Additionally, the loss of miR-501-3p also resulted in sociability, memory, and sensorimotor gating disruptions in male mice. The rescue of miR-501-3p reversed the above findings. 

Further results showed that miR-501-3p directly binds to the 3′UTR of Grm5, a gene that encodes glutamate metabotropic receptor 5 (mGluR5). mGluR5 is involved in neural network activity regulation and N-methyl-d-aspartate receptors (NMDAR) function and plasticity. The study shows that miR-501-3p downregulates Grm5 expression and thus decreases mGluR5. To evaluate the effects of mGluR5 in miR-501-3p knockout mice, they inhibited mGluR5 using MTEP and found that this improved impaired social preference, social novelty recognition, and novel object recognition. Finally, they found that mGluR5 works with NMDAR to produce increases in excitatory synaptic transmission in miR-501-3p knockout mice, which shows their necessity in the observed social and memory impairments. Overall, this study’s findings suggest that miR-501-3p could play a role in the pathogenesis of schizophrenia by modulating mGluR5 [79]. 

### 7.2. Cognition and Memory

Cognitive deterioration is one of the most significant symptoms of AD. Visualizing AD as a continuum, patients will go from normal cognition to mild cognitive impairment, and eventually, their cognition will continue to decline as they progress through the later stages of AD [80]. Recent studies have linked miR-501-3p to cognition. 

Gullett et al. (2020) studied miRNAs as predictive factors in the cognition of healthy older adults. MiR-501-3p was found to be one of the top-ranked predictors for fluid, crystallized, and overall cognition performance. The Montreal Cognitive Assessment (MoCA) was one of the tests used to assess cognition in the participants. The study found that miR-501-3p was negatively correlated with the MoCA score, which contrasted with an earlier study [81] that found a positive correlation between miR-501-3p and MMSE performance, which is another test that measures cognitive function. Despite the contrasting findings, these results from two different studies suggest that miR-501-3p plays a role in cognition [82]. 

Goldberg et al. (2021) delved deeper into the connection between miR-501-3p and cognition. Using an exercise model of memory-enhanced rats, the study showed an upregulation of miR-501-3p in rats after exercise. Since exercise has already been previously shown to improve memory function in humans, these findings suggest that miR-501-3p is upregulated in improved memory states. Furthermore, inhibition of miR-501-3p resulted in the upregulation of genes involved in neuronal cell death, autophagy, hypoxia, and ER stress and the downregulation of genes involved in structural and functional synaptic plasticity. The role of miR-501-3p was further characterized by the decreased dendritic spine density, number of mature synapses, and neuronal network activity observed after inhibition of the miRNA. Overall, these results indicate the importance of miR-501-3p in regulating synaptic integrity and synapse number. Hara et al. (2017) found downregulation of miR-501-3p in AD. Perhaps these findings hint that exercise could be used as a preventative measure or in the amelioration of AD [83]. 

Toyoma et al. (2018) proposed a specific mechanism for the role of miR-501-3p in cognitive impairment. VD is dementia caused by impaired cerebral perfusion. The study used a VD model in mice by performing bilateral common carotid-artery occlusion stenosis (BCAS) surgery. TNFα was elevated after BCAS surgery, which led to decreased claudin-5, ZO-1, and Occludin expression. These three proteins are key tight junction proteins that play a role in maintaining the blood–brain barrier (BBB). BBB disruption is one of the mechanisms thought to contribute to cognitive impairment in VD. Analysis of miRNA expression levels 48 h post BCAS surgery showed an upregulation of miR-501-3p in white matter endothelial cells. When mice that underwent BCAS surgery were treated with anti-miR-501-3p, the BCAS-induced decrease in ZO-1 was reversed. TNFα levels were unchanged in anti-miR-501-3p-treated BCAS mice when compared to BCAS mice that were not treated with anti-miR-501-3p, suggesting that miR-501-3p plays a role in the downregulation of ZO-1 due to TNFα. Additionally, anti-miR-501-3p treatment significantly ameliorated spatial working memory deficits and prevented white matter lesions. Overall, this study’s findings suggest that the TNFα-miR-501-3p-ZO-1 axis plays a role in hypoperfusion-induced working memory and white matter deficits due to BBB disruption and that therapy involved in targeting miR-501-3p could be beneficial in VD patients [84]. 

### 7.3. Alzheimer’s Disease

In 2017, Hara et al. introduced miR-501-3p as a novel serum biomarker for AD. The study found that miR-501-3p expression was downregulated in the serum but elevated in the brains, specifically the temporal cortex, of AD patients. Analysis at different stages of AD progression showed that miR-501-3p levels increased in the brain and decreased in the serum as the disease worsened. Overexpression of miR-501-3p was found to affect 208 significant genes, 128 of which were downregulated. Using TargetScan 7.1, they found that miR-501-3p was directly bound to 71/123 of the downregulated annotated genes (only 123 of the 128 downregulated genes were annotated in TargetScan 7.1). Genes that were downregulated were related to DNA replication and the mitotic cell cycle. The paper also concludes inappropriate re-entry into the cell cycle leads to apoptotic cell death, which precedes the development of senile plaques and NFTs [81].

Hu et al. (2015) explains one of the mechanisms of miR-501-3p in synaptic function. α-Amino-3-hydroxy-5-methyl-4-isoxazolepropionic acid receptors (AMPARs) are ionotropic glutamate receptors that determine the strengths of synapses. Previous studies have linked decreases in AMPARs and synaptic loss with AD [85]. Normally, NMDARs, which are metabotropic glutamate receptors, regulate AMPARs. This process is essential in determining the strength of synaptic connectivity during learning and memory. The study found that miR-501-3p is necessary for the NMDAR-mediated changes in AMPAR expression. MiR-501-3p is upregulated by NMDAR stimulation through the NMDAR subunit GluN2A. The miRNA then directly binds to the 3′UTR of *Gria1*, a gene that codes for the GluA1 subunit of AMPAR, and decreases GluA1 expression. The downregulation of GluA1 expression by miR-501-3p occurs locally in the dendrites and is vital for long-lasting spine remodeling [86]. As previously discussed, miR-501-3p was found to be upregulated in the brains of AD patients, and its expression increased as AD progressed. The findings from Hu et al. could provide a possible mechanism by which miR-501-3p participates in the pathogenesis of AD. Figure 4 describes the molecular mechanism of miR-501-3p in synapse dysfunction in AD.

## 8. MiR-502-3p and Alzheimer’s Disease

Previous studies have analyzed the differential expressions of miRNAs in the serum and brain of AD patients [76,81]. Synaptosomes contain the molecular machinery necessary to undergo synaptic function, so the miRNAs present in synaptosomes could be essential for synapses to function correctly. Our most recent study focused on synaptosomal miRNAs and their potential roles in the synaptic dysfunction that occurs in AD [29]. 

Using postmortem brains, we prepared synaptosomes of both AD cases and unaffected controls (UC). Analysis of miRNAs in AD synaptosomes vs. UC synaptosomes revealed several miRNAs that were significantly upregulated in AD synaptosomes compared to UC synaptosomes, including miR-502-3p. Additionally, we compared synaptosomal miRNA expression among AD samples in different Braak stages. We found that miR-501-3p, miR-502-3p, miR-877-5p, and miR-103a-3p were significantly different as AD progressed, suggesting that these miRNAs may be involved in AD progression [29]. 

One of the findings from an in silico Ingenuity^®^ Pathway Analysis (IPA) of the deregulated synaptosomal miRNAs in AD vs. UC samples showed that the miR-500 family, including miR-501-3p, was significantly linked to several biological processes and disorders. Further analysis showed that miR-501-3p was significantly involved in GABAergic synapse function. The *GABRA1* gene is a potential common target of miR-501-3p and miR-502-3p. Gene ontology enrichment analysis of miR-502-3p showed that the miRNA is significantly involved in external stimuli and nervous system development, and its most significant cellular component was the GABAergic synapse [29].

Previous studies have reported dysfunctional GABAergic synapses in AD. Our study supports this as we found reduced *GABRA1* levels in AD synaptosomes. Both IPA and Gene Ontology Enrichment show strong links between miR-501-3p and miR-502-3p with the GABAergic synapse pathways. It is possible that these two miRNAs could regulate the GABAergic pathway by negative modulation of *GABRA1* expression, leading to synaptic dysfunction seen in AD. This is further supported by our findings of increasing miR-501-3p and miR-502-3p expression with progressing Braak stages [29]. Figure 4 demonstrates the molecular mechanism of miR-502-3p in synapse dysfunction in AD.

Our initial findings of the relationship between miR-501-3p and miR-502-3p with AD pathology suggest that targeting miR-501-3p and miR-502-3p could be beneficial in ameliorating AD progression by preventing GABAergic synapse dysfunction. The miR-502-3p agomiRs mediated overexpression of miR-502-3p reduced *GABRA1* levels and reduced GABA current in the mouse primary hippocampal neurons. Meanwhile, the antagomiRs mediated suppression of miR-502-3p levels significantly increased the *GABRA1* levels as well as the GABA current [87]. Therefore, miR-502-3p plays a crucial role in regulating GABAergic synapse function in AD. However, further studies are necessary to characterize the specific molecular mechanisms of miR-501-3p and miR-502-3p in the context of synaptic functions. Furthermore, these miRNAs should also be addressed in other neurological disorders. IPA of miRNA-501-3p and other miRNAs with a similar seed (AUGCACC) shows involvement in multiple neurological functions. These include the function of GABAergic synapse, LMNA-related congenital muscular dystrophy, pain behavior, thermal hyperalgesia, mechanical allodynia behavior, and differentiation of muscle cell lines. 

In the laboratory, we are able to detect these miRNAs in various samples, such as AD postmortem brains, AD mouse brains, and cells by using real-time quantitative reverse transcription PCR (qRT-PCR), Northern blot, fluorescence in situ hybridization (FISH), microarray analysis, and miRNA sequencing [11,88]. For a living patient, biofluid or plasma may be analyzed with qRT-PCR or microarray analysis to determine miR-502-3p and miR-501-3p levels. These miRNA levels would serve as biomarkers of certain diseases. 

## 9. Conclusions

The purpose of this review was to comprehensively examine the roles of miR-502-3p across a variety of human diseases, including human cancers and neurodegenerative diseases. As discussed above, miR-502-3p has been heavily linked with several human cancers, but its relationship with neurodegenerative disorders is still fairly unexplored. IPA showed possible links between miR-502-3p and miR-501-3p with different neurological functions and disorders. Recently, we reported novel findings of miR-502-3p in the synaptosomes of AD patients and characterized its possible role in GABAergic synapse dysfunction. Further studies must be performed to reveal the molecular mechanisms of miR-502-3p and miR-501-3p in Aβ and p-tau mediated GABAergic and Glutaminergic synapse dysfunction in AD and other neurological diseases. We hypothesize that miR-502-3p and miR-501-3p can be a promising therapeutic target in the future to slow down the progression of AD. 

## Figures and Tables

**Figure 1 pharmaceuticals-16-00532-f001:**
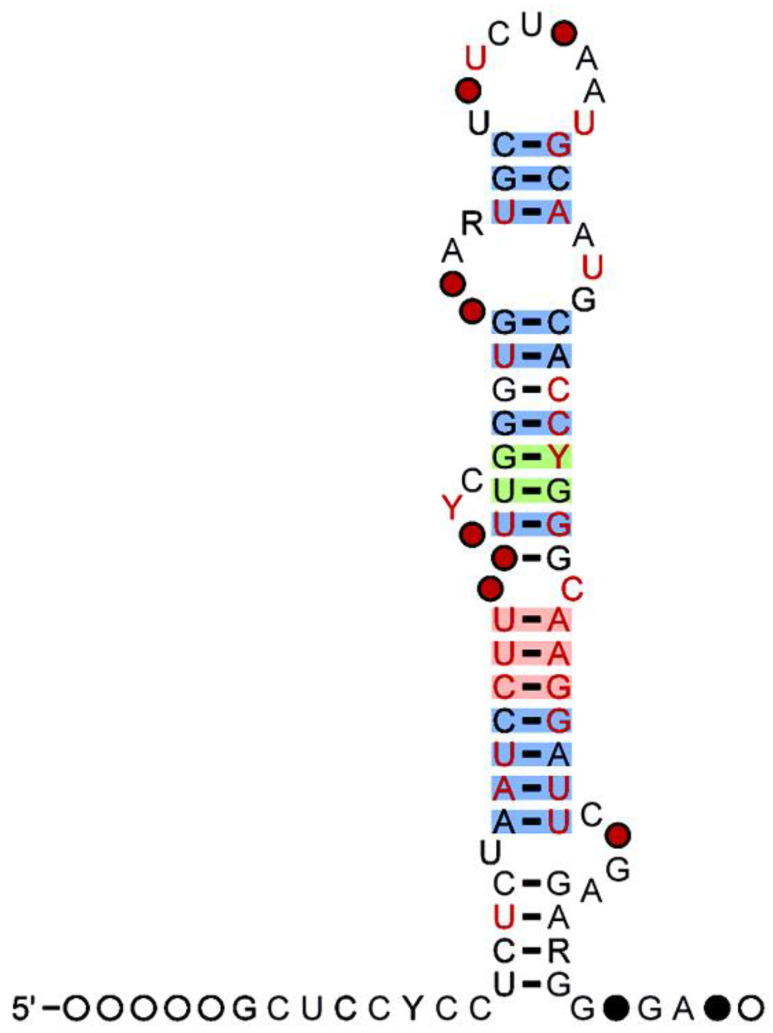
Secondary structure of miR-500 family showing presence of G-C Nucleotide. The 97 percent (%) of G-C nucleotide observed no mutations (Red). The 90 percent (%) of G-C nucleotide conveying mutations (Green) and the 75 percent (%) G-C nucleotide with compatible mutations (Source: useast.ensembl.org) (accessed on 25 February 2023).

**Figure 2 pharmaceuticals-16-00532-f002:**
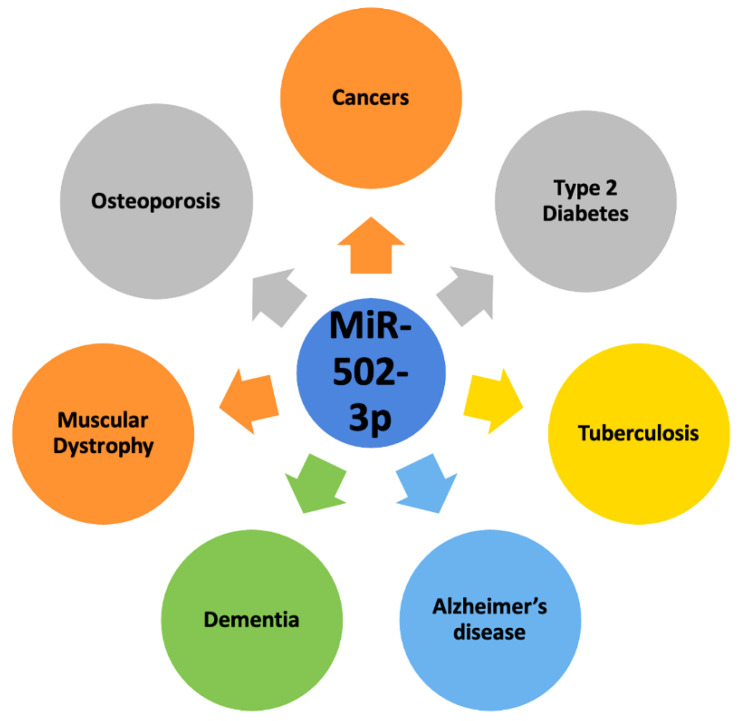
Involvement of miR-502-3p in most common human diseases. MiR-502-3p is involved in cancers, type 2 diabetes, tuberculosis, osteoporosis, muscular dystrophy, dementia, and Alzheimer’s disease.

**Figure 3 pharmaceuticals-16-00532-f003:**
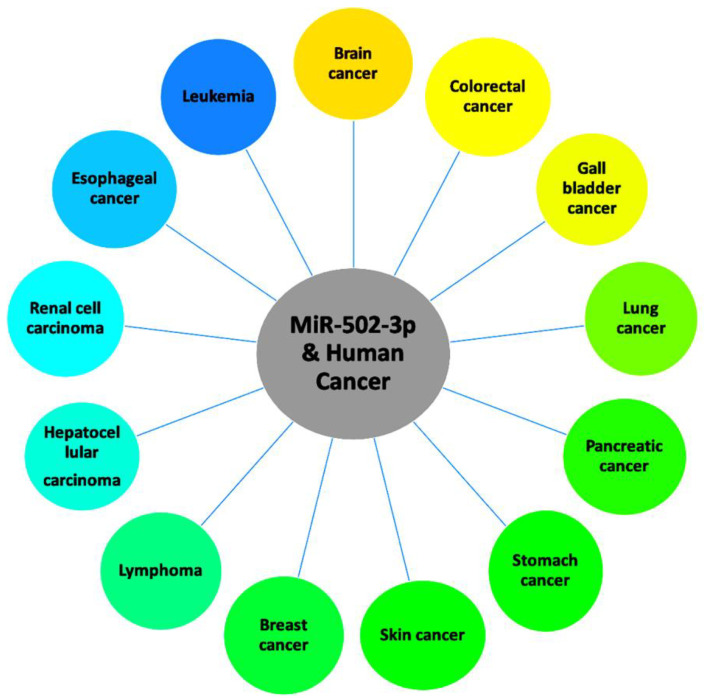
Roles of miR-502-3p in various human cancers.

**Figure 4 pharmaceuticals-16-00532-f004:**
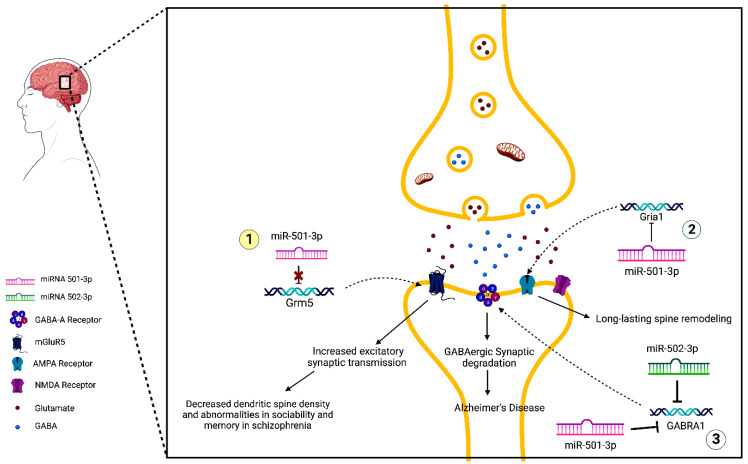
MiR-502-3p- and miR-501-3p-mediated molecular mechanisms of synaptic dysfunction in AD. (1) MiRNA-501-3p binds to the 3′UTR of Grm5 and inhibits its expression. In schizophrenia, miRNA-501-3p is downregulated, causing increased expression of Grm5 and mGluR5, the receptor for which it codes. This leads to increased excitatory synaptic transmission, which eventually results in the loss of total dendritic spine density and abnormalities in sociability and memory. (2) NDMAR-mediated changes in AMPAR expression occur via a miRNA-501-3p pathway. MiRNA-501-3p binds to the 3′UTR of Gria1 and inhibits its expression. This causes decreased production of the GluA1 subunit of AMPARs. This process is responsible for long-lasting spine remodeling. (3) Both miRNA-501-3p and miRNA-502-3p negatively modulate GABRA1 expression, leading to a decrease in GABA-A receptors. This may be a potential mechanism for the synaptic degradation of GABAergic synapses that lead to AD.

**Table 1 pharmaceuticals-16-00532-t001:** Details of miRNA-502-3p in various human diseases, their status, target genes, signaling pathways, and study purpose.

MiRNA-502-3p and Human Diseases					
Disease	miR-502-3p	Target Gene	Signaling Pathway	Purpose	Reference
Osteoporosis	Downregulated			Biomarker	Zhang et al., 2021
Diabetes Type 2	Upregulated			Therapeutic	Montastier et al., 2019
Tuberculosis	Upregulated	ROCK1	TLR4/NF-κB	Detrimental	Liu et al., 2021
Lamin A/C-associated Muscular Dystrophy	Upregulated				Sylvius et al., 2011
**MiRNA-502-3p and Human Cancers**					
**Disease**	**miR-502-3p**	**Target Gene**	**Signaling Pathway**	**Purpose**	**Reference**
Chronic Lymphocytic Leukemia	Upregulated		IL-4 pathway	Therapeutic	Ruiz-Lafuente et al., 2015
Invasive Pituitary Adenoma	Downregulated	KMT5A	LINC00473/miR-502-3p/KMT5A Axis		Li et al., 2021
Colorectal Cancer	Upregulated	OLFM4		Detrimental	Sugai et al., 2022
Gallbladder Cancer	Downregulated	SET	LncRNA-HGBC/miR-502-3p/SET/AKT Axis		Hu et al., 2019
Interstitial Lung Abnormality	Upregulated		p53 Signaling Pathway	Biomarker	Ortiz-Quintero et al., 2020
Lung Adenocarcinoma	Upregulated	MUC4		Detrimental	Subat et al., 2018
Pulmonary Ground-Glass Nodules	Upregulated			Prognostic	Zhang et al., 2019
Pancreatic Cancer	Downregulated	Various			Tan et al., 2018
Stomach Cancer	Downregulated	OLFM4	circ-RPL15/miR-502-3p/OLFM4/STAT3		Kim et al., 2021
Merkel Cell Carcinoma	Upregulated				Ning et al., 2014
Conjunctival Malignant Melanoma	Upregulated				Larsen et al., 2016
Triple Negative Breast Cancer	Upregulated				Zhang et al., 2019
Follicular Lymphoma	Upregulated				Wang et al., 2012
Hepatocellular Carcinoma				Detrimental	Liu et al., 2022
Renal Cell Carcinoma	Downregulated			Diagnostic	Wach et al., 2013
Esophageal Adenocarcinoma	Downregulated				Fassan et al., 2020
**MiRNA-502-3p and Neurodegenerative Disorders**					
**Disease**	**miR-502-3p**	**Target Gene**	**Signaling Pathway**	**Proposed Benefit**	**Reference**
Frontotemporal Dementia	Downregulated			Biomarker	Grasso et al., 2019
Vascular Dementia	Upregulated			Biomarker	Prabhakar et al., 2017
Alzheimer’s Disease	Downregulated			Biomarker	Satoh et al., 2015
Alzheimer’s Disease	Upregulated (Synapse)	GABRA1	GABAergic synapse	Therapeutic	Kumar et al., 2022
**MiRNA-501-3p and Neurodegenerative Disorders**					
**Disease**	**miR-501-3p**	**Target Gene**	**Signaling Pathway**	**Proposed Benefit**	**Reference**
Schizophrenia	Downregulated	Grm5		Therapeutic	Liang et al., 2022
Cognition	Negatively Correlated			Predictive	Gullett et al., 2020
Memory	Upregulated			Therapeutic	Goldberg et al., 2021
Vascular Dementia	Upregulated	ZO-1	TNFα/miR-501-3p/ZO-1 axis	Therapeutic	Toyama et al., 2018
Alzheimer’s Disease	Downregulated (serum), Upregulated (brain)				Hara et al., 2017
AMPAR Degradation	Upregulated	Gria1			Hu et al., 2015
Alzheimer’s Disease	Upregulated (Synapse)	GABRA1		Therapeutic	Kumar et al., 2022

## Data Availability

Not applicable.

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
