# Peer review of "Role of MicroRNA-502-3p in Human Diseases"

_pharmaceuticals, 2023, doi:10.3390/ph16040532_

Round 1

Reviewer 1 Report

1. The article is well-written and highly logical, with no significant grammar errors or language issues. However, it extensively discusses the role of miR-502-3p in various diseases, especially cancer, which is not relevant to the main topic of the article. Therefore, it should focus on exploring the role of miR-502-3p in neurodegenerative diseases, particularly Alzheimer's disease, rather than discussing other diseases such as cancer. Alternatively, the topic should be changed to better fit the content of the article.

2. The author mentioned in the abstract that "MiR-502-3p has been previously characterized in a variety of human diseases, including human cancers" but the main focus of this article is on Alzheimer's disease, which is not a human cancer.

3. "The most common causes of synapse dysfunction are amyloid beta, hyperphosphorylated tau, and microglia activation. "The sentence could be rewritten for greater clarity: "The most common causes of synapse dysfunction in AD are amyloid beta, hyperphosphorylated tau, and microglia activation."

4. "Current review focuses on the miR-502-3p and their potential roles" should be "Current review focuses on miR-502-3p and its potential roles".

5. The author discusses the importance of various miRNAs in AD in the section "2. MicroRNAs," but it is not clear why miR-502-3p is the most noteworthy role among many different miRNAs. It is necessary to increase the explanation of the importance of miR-502-3p to highlight the significance of this review article.

6. In the sentence "MiR-501-3p and miR-502-3p are very close members with single base sequence difference, however their seed sequence is similar," "single base sequence difference" should be changed to "a single base sequence difference."

7. "Table 1 needs improvement. The author's classification could be based on disease type, and focus on the exploration of miRNA-502-3p, which would be more relevant to the main topic of this article."

Author Response

Comments and Suggestions for Authors – Reviewer 1

We appreciate the reviewers for the careful checking of our manuscript and their thoughtful suggestions. The reviewers liked the topic and contents of the manuscript, however, there are some concerns raised by the reviewers to improve the quality and overall presentation of the manuscript. We addressed all the comments in the revised manuscripts highlighted in yellow and details of each comment are given below pint by point-

Comment 1. The article is well-written and highly logical, with no significant grammar errors or language issues. However, it extensively discusses the role of miR-502-3p in various diseases, especially cancer, which is not relevant to the main topic of the article. Therefore, it should focus on exploring the role of miR-502-3p in neurodegenerative diseases, particularly Alzheimer's disease, rather than discussing other diseases such as cancer. Alternatively, the topic should be changed to better fit the content of the article.

Response. We appreciate the reviewer’s suggestions. We changed the title of the article to better fit the content of the article.

Comment 2. The author mentioned in the abstract that "MiR-502-3p has been previously characterized in a variety of human diseases, including human cancers" but the main focus of this article is on Alzheimer's disease, which is not a human cancer.

Response. We appreciate the reviewer’s suggestions. We changed the title of the article to better fit the content of the article.

Comment 3. "The most common causes of synapse dysfunction are amyloid beta, hyperphosphorylated tau, and microglia activation. "The sentence could be rewritten for greater clarity: "The most common causes of synapse dysfunction in AD are amyloid beta, hyperphosphorylated tau, and microglia activation."

Response. We appreciate the reviewer’s suggestions. We changed the sentence as suggested to provide greater clarity.

Comment 4. "Current review focuses on the miR-502-3p and their potential roles" should be "Current review focuses on miR-502-3p and its potential roles".

Response. Our sincere apology for any grammatical errors. This sentence has been removed and replaced.

Comment 5. The author discusses the importance of various miRNAs in AD in the section "2. MicroRNAs," but it is not clear why miR-502-3p is the most noteworthy role among many different miRNAs. It is necessary to increase the explanation of the importance of miR-502-3p to highlight the significance of this review article.

Response. We appreciate the reviewer’s suggestions. We have expanded the section to expand on the importance of miR-502-3p as it relates to this review article.

Comment 6. In the sentence "MiR-501-3p and miR-502-3p are very close members with single base sequence difference, however their seed sequence is similar," "single base sequence difference" should be changed to "a single base sequence difference."

Response. Our sincere apology for any grammatic errors. This sentence has been revised as suggested.

Comment 7. "Table 1 needs improvement. The author's classification could be based on disease type, and focus on the exploration of miRNA-502-3p, which would be more relevant to the main topic of this article."

Response. We appreciate the reviewer’s suggestion. As suggested, the miR-502-3p classification is based on the disease type in the table.

Reviewer 2 Report

In this manuscript, Devara et al. provided a comprehensive review of mir-502-3p, trying to focus on Alzheimer’s disease. The authors focused on multiple aspects of the related areas, including miRNA, mir-500 family, mir-502-3p, human diseases, etc. This information is very useful for understanding the disease functions of mir-502-3p, which may serve as new therapeutic targets for these diseases. I have a few comments on this manuscript:

1, It would be beneficial to illustrate the techniques to detect these miRNAs, and how would these technologies be applied to clinical?

2, The title says the focus is Alzheimer’s disease; however, lots of discussions are related to diverse disease areas such as cancer. I would suggest making the title broader to better summarize the review.

3, I wonder if researchers have predicted the target genes of mir-502-3p? It would be useful if the authors include that, which will provide a list of genes for further validation. In Table 1, only a few target genes of mir-502-3p have been confirmed, which seems to be underrepresented.

4, What’s the expression profile of mir-502-3p in human tissues? Does it have higher expression in specific tissues?

Overall, I believe this is a good review paper demonstrating the diverse function of mir-502-3p in human diseases. I suggest a minor revision and look forward to the revised manuscript.

Author Response

We appreciate the reviewers for the careful checking of our manuscript and their thoughtful suggestions. The reviewers liked the topic and contents of the manuscript, however, there are some concerns raised by the reviewers to improve the quality and overall presentation of the manuscript. We addressed all the comments in the revised manuscripts highlighted in yellow and details of each comment are given below pint by point-

In this manuscript, Devara et al. provided a comprehensive review of mir-502-3p, trying to focus on Alzheimer’s disease. The authors focused on multiple aspects of the related areas, including miRNA, mir-500 family, mir-502-3p, human diseases, etc. This information is very useful for understanding the disease functions of mir-502-3p, which may serve as new therapeutic targets for these diseases. I have a few comments on this manuscript:

Comment 1. It would be beneficial to illustrate the techniques to detect these miRNAs, and how would these technologies be applied to clinical?

Response. We appreciate the reviewer’s suggestions. We have included an extra paragraph to list the most common techniques used to detect the miRNAs.

Comment 2. The title says the focus is Alzheimer’s disease; however, lots of discussions are related to diverse disease areas such as cancer. I would suggest making the title broader to better summarize the review.

Response. We appreciate the reviewer’s suggestions. We changed the title of the article to better fit the content of the article.

Comment 3. I wonder if researchers have predicted the target genes of mir-502-3p? It would be useful if the authors include that, which will provide a list of genes for further validation. In Table 1, only a few target genes of mir-502-3p have been confirmed, which seems to be underrepresented.

Response. We appreciate the reviewer’s suggestions. Unfortunately, the authors of the reviewed papers did not include a list of genes for further validation because some of the findings were produced due to a wide screening for miRNA expression changes. In those papers, miR-502-3p was mentioned, but it was not focused upon. We conducted further analysis and provided a new supplemental table that lists the top 100 predicted targets for miR-501-3p and miR-502-3p.

Comment 4. What’s the expression profile of mir-502-3p in human tissues? Does it have higher expression in specific tissues?

Response. We appreciate the reviewer’s suggestions. We have added a new portion in the “3. MiR-500 Family” section about the expression of miR-502-3p in human tissues.

Overall, I believe this is a good review paper demonstrating the diverse function of mir-502-3p in human diseases. I suggest a minor revision and look forward to the revised manuscript.